# An Efficient Dummy-Based Location Privacy-Preserving Scheme for Internet of Things Services

**Yongwen Du [1], Gang Cai [1], Xuejun Zhang [1],\*, Ting Liu [1] and Jinghua Jiang [2]**

[1]   School of Electronic and Information Engineering, Lanzhou Jiaotong University, Lanzhou 730000, China
[2]   Baidu Inc.Baidu Building, 10 Shangdi 10th Street, Haidian District, Beijing 100000, China
\*   Correspondence: xuejunzhang@mail.lzjtu.cn

**Abstract:** With the rapid development of GPS-equipped smart mobile devices and mobile computing, location-based services (LBS) are increasing in popularity in the Internet of Things (IoT). Although LBS provide enormous benefits to users, they inevitably introduce some significant privacy concerns. To protect user privacy, a variety of location privacy-preserving schemes have been recently proposed. Among these schemes, the dummy-based location privacy-preserving (DLP) scheme is a widely used approach to achieve location privacy for mobile users. However, the computation cost of the existing dummy-based location privacy-preserving schemes is too high to meet the practical requirements of resource-constrained IoT devices. Moreover, the DLP scheme is inadequate to resist against an adversary with side information. Thus, how to effectively select a dummy location is still a challenge. In this paper, we propose a novel lightweight dummy-based location privacy-preserving scheme, named the enhanced dummy-based location privacy-preserving(Enhanced-DLP) to address this challenge by considering both computational costs and side information. Specifically, the Enhanced-DLP adopts an improved greedy scheme to efficiently select dummy locations to form a k-anonymous set. A thorough security analysis demonstrated that our proposed Enhanced-DLP can protect user privacy against attacks. We performed a series of experiments to verify the effectiveness of our Enhanced-DLP. Compared with the existing scheme, the Enhanced-DLP can obtain lower computational costs for the selection of a dummy location and it can resist side information attacks. The experimental results illustrate that the Enhanced-DLP scheme can effectively be applied to protect the user's location privacy in IoT applications and services.

**Keywords:** Internet of Things; location-based service; dummy location selection; location privacy

## 1. Introduction

In recent years, with the rapid development of smart mobile devices and mobile communication technologies, Internet of Things (IoT) services have emerged in our daily life [1]. According to the annual report of the International Data Corporation, the IoT market will exceed $7 trillion by 2020 [2]. Many potential IoT applications rely on location-based services (LBS) or LBS applications, such as GPS navigators, information retrieval, and context-aware mobile applications, etc. [3]. To benefit from these IoT applications, users can obtain the required geographic location services by sending their current location or point of Interest (POI) to the untrusted LBS providers (LSPs), for instance, querying to find the nearest shopping center [4].

Although LBS provides enormous convenience and benefits to IoT users, it also raises serious privacy concerns because location information is collected by untrusted or malicious LSPs [4]. By analyzing users' location information, untrusted or malicious LSPs can infer users' personal information by associating a user's identity with queried locations and interests [4], such as their home addresses, sexual preference and health conditions, etc. In order to alleviate the location privacy

concerns in IoT services and applications, many privacy protection schemes have been proposed recently. Most of the schemes are based on location k-anonymity technology. The k-anonymity technique is one of the most important techniques to protect users' location privacy in LBS; it can ensure that a user's location is identified with a probability at less than or equal to $1/k$. The dummy location is a fake query location that is similar enough to the user's location so that the attacker can not distinguish which is the user's real location from the anonymous set. Dummy locations are widely used to realize location k-anonymity due to the following advantages: (1) they do not rely on complex encryption and decryption schemes, (2) they effectively save computing resources of the IoT device, (3) they achieve better trade-offs between LBS privacy and utility.

The dummy-based location privacy-preserving (DLP) scheme is widely used in IoT services [5–10], which focus on how to effectively select reasonable dummy locations and avoid having the real location distinguished by LSPs. However, these schemes do not fully consider the side information [5,7,10]. In reality, some attackers may have such side information, and these dummy-based location selection schemes are inadequate to preserve IoT users' privacy. Some schemes do consider the side information available to attackers [6,8,9], and these schemes are based on the entropy metric [11]. However, the computational cost of these schemes is very expensive, and this is not suitable for resource-constrained IoT devices. To the best of our knowledge, how to select reasonable dummy locations is still a challenge, especially for a data-driven IoT service.

In this paper, we propose an enhanced dummy-based location privacy-preserving scheme (Enhanced-DLP), which aims for practical efficiency for IoT users and devices with strong location privacy protection. The basic idea of Enhanced-DLP is that the dummy locations are selected to build an optimal anonymous set for IoT user by considering the side information that is available to an adversary. Specifically, the Enhanced-DLP scheme adopts an enhanced greedy scheme when constructing the anonymous set. The dummy location that is most similar with the current anonymous set will be added to the anonymous set. Comparing our Enhanced-DLP with previous state-of-the-art schemes, the experimental results illustrate that our Enhanced-DLP scheme can reduce the computational costs while preventing location privacy breaches from adversaries with side information. The main contributions of this paper are as follows:

1.  The dummy-based privacy protection scheme is a privacy protection scheme without a third trusted party, and is more suitable for the IoT environment. In this paper, we propose an enhanced greedy manner to improve the existing dummy-based location selection scheme (such as DLP) and improve the efficiency of dummy location generation. Compared with the DLP scheme, the dummy location generation time of our scheme is reduced by 68.14%.
2.  The Enhanced-DLP scheme enhances the user's privacy protection, making it difficult for an attacker to distinguish the user's real location from the anonymous set. Compared with the DLP scheme, the probability of our scheme revealing the user's real position is reduced by 33.63% under the same attack conditions.

## 2. Preliminaries

### 2.1. Side Information

Side information [10] is part of the background knowledge and is usually related to other information of the user at a specific time and space. The side information also is the probability of a query at a certain location or time, or some information related to the query semantics, such as the user's gender, age, historical location, and social status. The side information in this paper is characterized by a mathematical model of the historical query probability of the locations. In particular, if no such historical log is available, a simple way to approximate the historical query probability is to count the number of POIs, as a user is likely to query for POI that reside around their location [12]. The historical query probability of a certain location is denoted as the ratio of the number of queries

submitted in the current location to the number of queries submitted in all locations, as shown in Equation (1).

$$p_i = \frac{number\ of\ queries\ submitted\ in\ location\ i}{number\ of\ queries\ submitted\ in\ all\ locations}. \tag{1}$$

*2.2. The Entropy-Based Privacy Measurement Method*

Entropy was first used to measure the degree of confusion of an anonymous set in [11]. In this paper, entropy is also used to measure the degree of privacy preservation for an anonymous set. It can be regarded as the uncertainty of recognizing the user's location from the dummy location set. The greater the entropy, the harder it is to recognize the user's location from the anonymous set. The entropy is calculated based on the historical query probability (denoted as $p_i$) for each location in the anonymous set, which can be summarized by Equation (2):

$$H = -\sum_{i=1}^{k} q_i \log_2 q_i \tag{2}$$

where $q_i = p_i / \sum_{i=1}^{k} p_i$ The sum of all $q_i$ $(i = 1, 2, 3, ..., k)$ is 1. This ensures that the historical query probability of all locations in the anonymous set is in the same range.

*2.3. Review the Dummy-Based Location Privacy-Preserving Scheme*

The DLP scheme is used to solve the problem that the traditional dummy location selection scheme has a higher computational cost when selecting the dummy location. According to the value of $k$, the DLP scheme needs to efficiently generate $k - 1$ dummy locations against side information that may be exploited by the attacker. The following steps show how it works:

1. A user needs to give the degree of anonymity $k$.
2. The DLP scheme needs to obtain the historical query probability of all locations and sort them in ascending order. Then, the DLP scheme calculates the number of locations (denoted as $N$) with the same historical query probability of the user's location (denoted as $p$). The anonymous set of the user is denoted as $C$.
3. If $N \geq k$, the DLP scheme arbitrarily selects $k - 1$ locations whose query probability is the same as $p$ from $N$ locations and user location $p$ to form an anonymous set $C$.
4. If $k > N \geq k/4$, the DLP scheme selects $N - 1$ locations with the same query probability as the user location $p$ from the sorted list and puts them into $C$. The DLP scheme selects $2k$ locations that are the most similar to the user's location $p$ as a candidate set (denoted as $S$). The DLP scheme uses the greedy scheme $k - N$ times to find $k - N$ dummy locations from $S$, and also puts them into $C$ to form the anonymous set.
5. If $N < k/4$, the DLP scheme chooses $2k - \varepsilon$ locations less than, and $2k - \omega$ locations greater than the real location to form a $4k - \omega - \varepsilon$ candidate set from the sorted list (denoted as $S_b$).Then, the DLP arbitrarily selects one location from $S_b$ and puts the location and the user's real location into $C$. The DLP scheme uses the greedy scheme $k - 2$ times to find $k - 2$ dummy locations and puts them into $C$ to form an anonymous set.
6. Finally, the optimal anonymous set $C$ is found.

Greedy scheme: A complex problem is divided into several steps, and each step can obtain a local optimal solution, so that the final result approximates the global optimal solution. In the Enhanced-DLP scheme, the basic idea of the greedy scheme is to find the optimal dummy location step by step from a candidate set. According to the measurement of the positional entropy, the dummy location selected in each step must ensure that the current anonymous set can obtain the max entropy. In the greedy scheme of the DLP scheme, each greedy choice needs to decide which location in the

candidate set can give the current anonymous set the maximum entropy, and finally achieve the optimal dummy location.

From the analysis of the DLP scheme, it can be seen that the DLP scheme is divided into three branches. In the first branch and the third branch, the scheme of generating the dummy location is already optimal. Our work is mainly to improve the scheme of generating the dummy location in the second branch by using the improved greedy scheme, which makes it possible to find the current optimal dummy location without traversing all the locations in the candidate set when generating the dummy location, in order to obtain a lower computational cost and lower attack probability. In addition the Enhanced-DLP can guarantee that the anonymous set has the same entropy as the anonymous set generated by the existing DLP scheme.

### 2.4. System Model

In reality, a large amount of IoT applications are based on LBS which are the key elements for implementing IoT location services (as shown in Figure 1). For instance, the location information of an object being transported can be accurately located through the LBS while in the process of transportation. The Enhanced-DLP scheme can be adapted to these scenarios in the IoT.

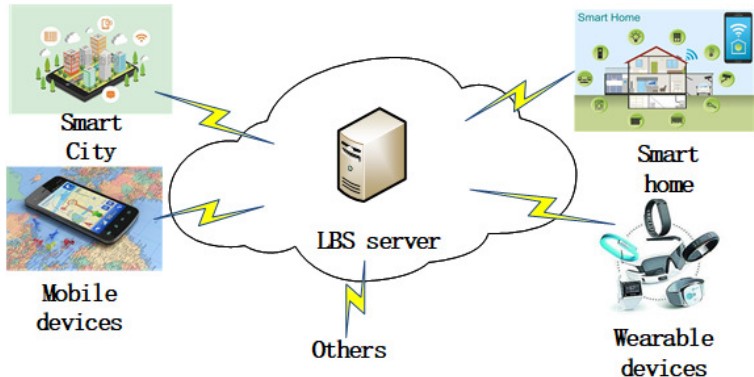

**Figure 1.** Location services and applications in the Internet of Things (IoT).

There are two main parts in the system model: the LBS server and IoT user devices, as shown in Figure 2.

- **The LBS server**: The LBS server receives the location service request from the user devices, and then searches the requested service in the LBS database and sends the search results back to the user. The LBS server contains various types of service databases and provides users with diverse services. Moreover, the server may also count the probability of each user's query at all locations when providing service information for each user according to the location.
- **The IoT user device**: The IoT user device is a typical part of the location service system, which usually obtains user location information through a GPS module embedded in the device (e.g., a smartphone). The IoT device obtains the corresponding high-quality location service by sending its high-precision location information to the LBS server. In order to prevent the user's location information from being used twice, the location information submitted to the LBS server should include the accurate location and dummy location, which ensures that the LBS server is not able to obtain the user's real location.

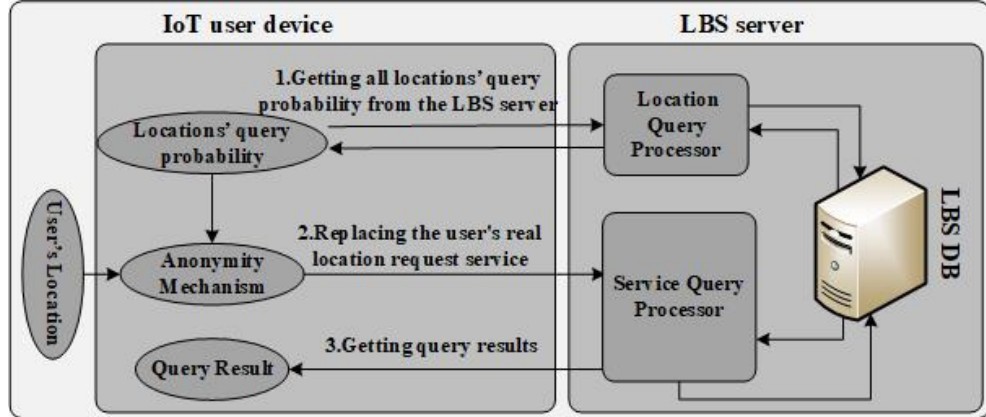

**Figure 2.** LSB privacy-protection model in IoT.

*2.5. Attack Model*

The ultimate goal of the malicious attacker is to find the user's real location information from the k anonymous set (dummy location set). The adversary may capture the LBS server to obtain all locations data and monitor the location queries submitted by the user. So, in this paper, the LBS server can be considered as an attacker. In addition, we consider that the privacy protection scheme is generally public, which is obtained by attackers.

## 3. Enhanced-DLP Scheme

In the domain of the IoT privacy-preservation, the dummy-based location scheme has been widely used for preserving the user's location privacy. For example, the DLS and DLP schemes are two well-known dummy-based location selection schemes for a data-driven IoT LBS, which consider the side information available to an adversary. However, the computational cost of these two schemes are very expensive, and the DLS is vulnerable to privacy breaches with Attack scheme for DLS(ADLS) attacks (This is a dummy-based location selection attack scheme proposed in reference [9]). Aiming at solving these problems, we present an Enhanced-DLP scheme which can efficiently select dummy locations to form an anonymous set. The Enhanced-DLP scheme selects dummy locations in an enhanced greedy manner for a tradeoff between computational cost and privacy requirements, which can effectively reduce the computational complexity of the dummy location selection and improve the level of privacy protection for the user.

*3.1. The Basic Idea of the Enhanced-DLP Scheme*

The goal of the Enhanced-DLP scheme is to efficiently select the dummy locations in an enhanced greedy manner by considering some side information that is available to an adversary. Namely, when constructing the $k$ anonymous set, we select the $k$ dummy locations with the most similar history query probability to the user's real location in an enhanced greedy manner. Our goal is to generate an anonymous set with sufficiently large entropy. For example, we suppose that there is a candidate set (denoted as $D$) of size $2k$. If the Enhanced-DLP has already selected $i$ (where $i < k$) locations from set $D$ as a set denoted as $C$. When selecting the $(i+1)^{th}$ location, the Enhanced-DLP needs to calculate the location with the maximum query probability (denoted as $p_{max}$) and the location with the minimum query probability (denoted as $p_{min}$) in the anonymous set $C$. Then the Enhanced-DLP finds two locations from candidate list $D$, where one is the location with the maximum query probability less than $p_{min}$ (denoted by $p_{min-max}$), and the other is the location with the minimum query probability greater than $p_{max}$ (denoted by $p_{max-min}$). Finally, the Enhanced-DLP needs to calculate $H(C, p_{min-max})$

and $H(C, p_{max-min})$ according to Equation (3). From Equation (4), the $p_{n+1}$ is not greater than the maximum of set $C$, and the $p_{n+1}$ is not less than the minimum of set $C$ as well.

$$H(C, p_{n+1}) = -\sum_{p_j \in C} \frac{p_j}{\sum_{p_j \in C} p_j + p_{n+1}} \log_2 \frac{p_j}{\sum_{p_j \in C} p_j + p_{n+1}}$$
$$- \frac{p_{n+1}}{\sum_{p_j \in C} p_j + p_{n+1}} \log_2 \frac{p_{n+1}}{\sum_{p_j \in C} p_j + p_{n+1}} \tag{3}$$

$$\min(C) \leq p_{n+1} = \exp\left(\frac{\sum_{p_j \in C} p_j \log_2 p_j}{\sum_{p_j \in C} p_j}\right) \leq \max(C) \tag{4}$$

where $p_j$ denotes the historical query probability at location $j$.

The Enhanced-DLP scheme's basic steps are as follows:

1.  In the beginning, a user sets the value of the privacy degree $k$ according to the privacy requirement. Then, the Enhanced-DLP obtains all the historical query probability of all locations on the LBS server and sorts them in ascending order according to the historical query probability (denoted as $S$). Let $p$ denote the query probability of the user's real location, and let $C$ denote the final anonymous set.
2.  The Enhanced-DLP scheme counts the number of locations that have the same historical query probability as $p$ in set $S$, which is denoted by $k_b$.
3.  If size $(k_b) \geq k$, then the Enhanced-DLP randomly selects $k-1$ locations that have the same historical query probabilities as the user's location $p$ from $k_b$ and puts them into $C$. Then, the Enhanced-DLP randomly inserts the user's real location $p$ into $C$.
4.  If $1 < size(k_b) < k$, the Enhanced-DLP scheme puts each location in $k_b$ and user's location $p$ into $C$. Then, the Enhanced-DLP selects $2k$-2 candidate locations (denoted as set $D$), in which $k-1$ locations are less than the $p$ and the other $k-1$ locations are greater than the $p$ in the set $S$.

    Otherwise, it goes to Step (7).
5.  The Enhanced-DLP finds the location maximum with query probability (denoted as $p_{max}$) and location with minimum query probability (denoted as $p_{min}$) from set $C$. The Enhanced-DLP finds two locations in set $D$, which is the location with the maximum query probability less than $p_{min}$ (denoted by $p_{min-max}$), and the other is the location minimum query probability being greater than $p_{max}$ (denoted by $p_{max-min}$). Then, the Enhanced-DLP needs to calculate $H(C, p_{max-min})$ and $H(C, p_{min-max})$ according to Equation (3) and add the location into set $C$, to attempt to achieve max entropy in set $C$.
6.  Repeat Step (5) while the size of anonymity set $C$ is less than $k$.
7.  If size $(k_b) \leq 1$, the Enhanced-DLP chooses $2k - \varepsilon$ locations less than $P$ and other $2k - \omega$ locations greater than the $p$ as $4k - \omega - \varepsilon$ candidate locations from the sorted set $S$ and denoted as $S_b$. Both $\varepsilon$ and $\omega$ are set by users based on their privacy requirements, and $\varepsilon$ is greater than $\omega$. Then, the Enhanced-DLP randomly gets one location from $S_b$ as a dummy location, and puts this location and the user's location $p$ into $C$.
8.  In order to meet the requirements of k-anonymity, the Enhanced-DLP selects other $k-2$ locations from $S_b$. For the $i^{th}$ location that Enhanced-DLP selected, The Enhanced DLP must make sure that $H_i$ is the largest of the remaining positions in set $S_b$.
9.  If the size of set $C < k$. Repeat Step (8).

    And then outputs the anonymous set $C$.

The detailed pseudo-code of the Enhanced-DLP is seen in Algorithm 1.

---

**Algorithm 1** Enhanced-DLP algorithm.

---

**Input:** The set of all locations with query probability $S$; users' location $p$.
**Output:** The optimal set of dummy locations $C$.
 1: Sort $S$ in ascending order;
 2: $k_b \leftarrow$ Select the locations which has the same query probability as user's real location $p$ from sorted set $S$.
 3: **if** size($k_b$)$\geq k$ **then**
 4:      $C \leftarrow$ Randomly select $k - 1$ dummy locations from $k_b$.
 5:      Randomly insert location $p$ into set $C$
 6: **end if**
 7: **if** $1 < size(k_b) < k$ **then**
 8:      $C \leftarrow C \cup k_b, S \leftarrow S / k_b$
 9:      $C \leftarrow C \cup p, S \leftarrow S / p$
10:      $D \leftarrow$ Choose $k - 1$ locations less than user's location $p$ and $k - 1$ locations greater than user's location $p$ in the sorted list $S$;
11:      **for** $i = 1; i < k - 2; i++$ **do**
12:          $p_{max} \leftarrow$ max(C)
13:          $p_{min} \leftarrow$ min(C)
14:          Find one location from set $D$, which is the location with maximum query probability being less than $p_{min}$ in set $D$, denoted as $p_{min-max}$;
15:          Find one location from set $D$, which is the location with minimum query probability being greater than $p_{max}$ in set $D$, denoted as $p_{max-min}$;
16:          **if** $H\left(C, p_{max-min}\right) > H(C, p_{min-max})$ **then**
17:             $C \leftarrow C \cup p_{max-min}, D \leftarrow D / p_{max-min}$;
18:          **else**
19:             $C \leftarrow C \cup p_{min-max}, D \leftarrow D / p_{min-max}$;
20:          **end if**
21:      **end for**
22: **else**
23:      $S_b \leftarrow$ Choose $4k - \omega - \varepsilon$ candidate locations, whose $2k - \omega$ locations less than user's location $p$ and $2k - \varepsilon$ locations greater than user's location $p$ in the sorted list $S$
24:      Randomly choose location $i$ from set $S_b$;
25:      $C \leftarrow H \cup i, S_b \leftarrow S_b / i$;
26:      **for** $j = 1; j \leq k - 2; j++$ **do**
27:          Choose one location $h$ from $S_b$, which makes sure that $H(C, h)$ is the maximum in set $S_b$;
28:          $C \leftarrow C \cup h, S_b \leftarrow S_b / h$;
29:      **end for**
30: **end if**
31: the optimal set $C$.

---

*3.2. Security Analysis*

●    Resistance to the Colluding Attack

**Proof.** A passive attacker who would like to obtain the user's location information may collude with other users or LBS servers. A colluding attack happens with a number of users that want to recognize the user's location information from the submitted $k$ locations by colluding each other. □

In our scheme, a user wants to protect his/her location privacy by way of choosing other locations with the same query probability as the user's location. Once an attacker compromises a user $U_1$, the rest of the $k - 1$ location information which has the same historical query probability as $U_1$ will be intercepted.The attacker can not recognize which one is the user's location from k locations because they have similar historical query probability. The only way is for the attacker to randomly select a location from the $k$ locations set to be the user's location. Therefore, the probability of identifying the user's location information from $k$ locations is $1/k$. Then, the user $U_2's$ LBS query is intercepted by the attacker who receives the user's location information. However, the probability to identify the user's location information from $k$ locations is still $1/k$. The reason is that the selected dummy locations of

user $U_1$ and $U_2$ have no correlations. The attacker identifies the user's real location from intercepted $k$ locations by way of random selection. Moreover, with a colluding group with more users, the attacker can only identify the user's real location randomly from intercepted $k$ locations. We can see that the probability to recognize the user's real location from the generated dummy location set is kept steady.

- Resistance to a Side Information Inference Attack

**Proof.** An active attacker wants to obtain the user's location privacy information according to the historical query probabilities of all locations and the current queries of users. In this paper, we assume that the LBS server mentioned in system model is an active attacker. □

In the Enhanced-DLP scheme, from Step (3) to Step (5), the LBS server can not distinguish the user's real location from $k$ locations, even if the attacker knows the historical query probability of all locations, since the selected $k$ locations all have the same historical query probability as the user's location. From Step (6) to Step (20), due to the selected $k$ locations having similar historical query probability, it is hard for the attacker to distinguish the user's real location from an anonymous set. The reason is that the Enhanced-DLP scheme can guarantee that the chosen dummy locations comprise an anonymous set with the largest entropy and enough dummy locations whose query probability are the same as the user's real location, thus the LBS server still cannot identify the user's real location from $k$ locations. From Step (21) to Step (29), as the chosen $k$ dummy locations have a similar historical query probability and the Enhanced-DLP ensures the uncertainty of the selection, the LBS server cannot identify the user's real location from $k$ locations.

## 4. Experimental Results and Analysis

In order to evaluate the performance of the Enhanced-DLP, we designed and implemented comprehensive experiments using the Python 3.5 language environment. All experiments were performed on an Intel i5 2.5-GHz machine with 12 GB RAM and the Windows 10 operating system. Moreover, the scheme run time is measured in terms of CPU time. In the experiment, the service map data of the LBS provider is divided into n × n cells and each cell has the same size for IoT users to submit queries. We set a historical query probability in these cells. We set two scenarios to evaluate the performance of our scheme.

**Scenario 1**: There are more than $k$ locations that have the same historical query probability as the user's real location. The Enhanced-DLP will select some dummy locations with the same historical query probability as the user's real location. The dataset is from the NA dataset [13] the North America road network, which contains 175,813 real POIs.

**Scenario 2**: Let the user be located in a cell such that there are a few (less than $k$ but greater than 1) cells which have the same probability as the user's current location. In this scenario, the Enhanced-DLP can choose enough dummy locations which have the same query probability to form a anonymous set for the user. The dataset is from an n × n grid map whose cell's historical query probability is randomly generated.

We have compared the performance of five schemes (i.e. DLS [8], DLP [9], GridDummy [14], CircleDummy [14], and Optimal) in terms of the run time and the privacy level under various degrees of anonymity. We also compare the probability of query recognition with DLS and DLP under Scenario 2.

Figure 3a,b illustrate the results of the comparison of the degree of privacy protection and run time with the Enhanced-DLP, DLS, DLP, Optimal, GridDummy and CircleDummy scheme in Scenario 1. From the Figure 3a, we can see that the entropy obtained by all schemes increases continuously when the value of k increases, indicating that the privacy protection capability of all schemes is increasing. The Enhanced-DLP, the DLP, the Optimal, and the DLS scheme have the largest entropy. The reason is that the Enhanced-DLP scheme adopts the enhanced greedy scheme to ensure that the final anonymous set entropy value is the largest. The GridDummy and CircleDummy scheme select dummy locations adjacent to the user's location based on their geographic location, so the GridDummy

and the GridDummy scheme cannot obtain the maximum entropy, and the obtained entropy is not stable. From the Figure 3b, we can see that the computational cost of the Enhanced-DLP and the DLP scheme increases with the growth of the value of *k*, but the run time of the Enhanced-DLP is 3.77% lower than the run time of the DLP. The reason is that the Enhanced-DLP adopts an enhanced greedy scheme which has a lower computational cost. In addition, the scheme of generating dumb positions in DLP and enhanced-DLP in this scenario is almost the same. Our work mainly reduces the computational cost of the scheme in the dataset of Scenario 2. The run time of the other schemes can be seen in Table 1 where the DLP and Enhanced-DLP schemes have a shorter time, while the DLS and CircleDummy run times are very long. The run times of CircleDummy and DLS rapidly increase as the value of *k* increases. The reason is that the DLS scheme select *k* dummy locations by the way of the enumeration scheme which has a high computational cost, and the CircleDummy scheme needs to calculate the angle and distance between all locations and the user location which also has a high computational cost.

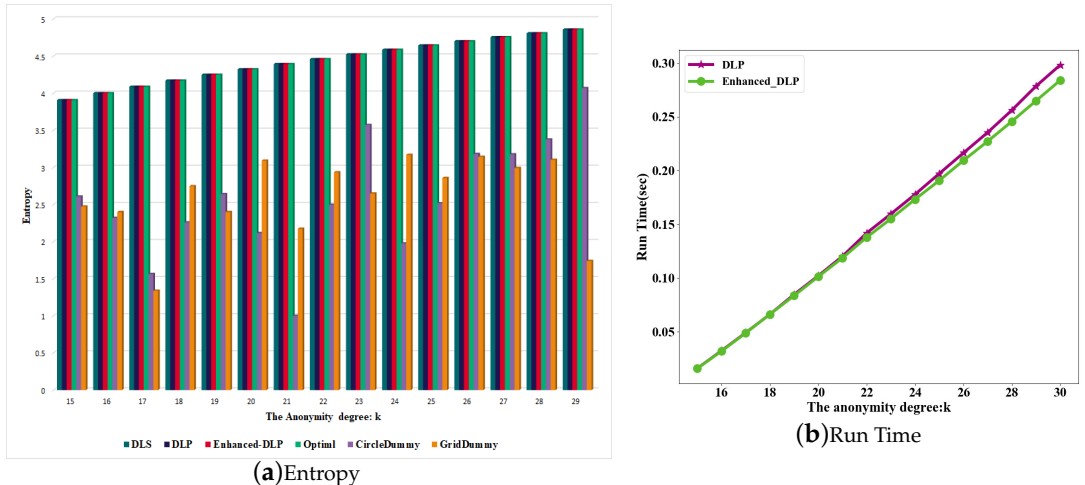

(**a**)Entropy      (**b**)Run Time

**Figure 3.** Scenario 1: Entropy and run time.

**Table 1.** Run times for Scenario 1.

| Scheme | Anonymity Degree: k | | | |
|---|---|---|---|---|
| | **k = 15** | **k = 20** | **k = 25** | **k = 29** |
| DLP | 0.019 s | 0.117 s | 0.217 | 0.298 s |
| DLS | 0.431 s | 3.184 s | 6.870 s | 10.553 s |
| CircleDummy | 1.236 s | 9.003 s | 19.316 s | 29.604 s |
| GridDummy | 0.00003 s | 0.0002 s | 0.0004 s | 0.0006 s |
| Enhanced-DLP | **0.018 s** | **0.116 s** | **0.215 s** | **0.295 s** |

Figure 4a,b illustrate the results of the comparison of entropy and run time in Scenario 2. In Figure 4a, the entropy obtained by all schemes increases continuously when the value of *k* increases. While the entropy of the Enhanced-DLP and the DLP scheme are the same, there are big differences in the computational costs in Figure 4b. With an increase in *k*, the run time of the DLP scheme increases rapidly but the Enhanced-DLP run time varies little. And the run time of the Enhanced-DLP is 68.14% lower than that of the DLP. The reason is that the DLP scheme adopts the greedy scheme to select $k-1$ dummy locations, which needs to calculate all the residual locations in candidate set to ensure the current set has the largest entropy. While the Enhanced-DLP scheme adopts the enhanced greedy manner, which only needs to calculate $p_{max-min}$ and $p_{min-max}$ to successively select $k-1$ dummy locations to ensure the largest entropy. Furthermore, the computational complexity of the DLP scheme (i.e., $O(n^2)$) is higher than the Enhanced-DLP scheme (i.e., $O(n)$), where *n* and *k* are the same. The experimental results

are also consistent with the results of our analysis. The run time of Enhanced-DLP is much lower than that of DLP.

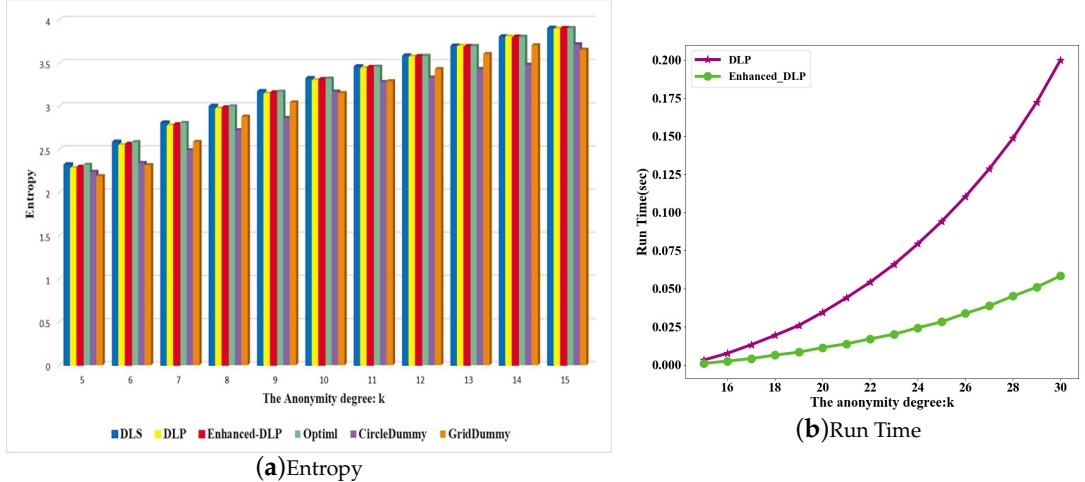

(**a**)Entropy

**Figure 4.** Scenario 2: Entropy and run time.

Figure 5 illustrates the probability of query recognition under an ADLS attack for three different scheme. As the value of k increases, the probability of query recognition of three schemes is gradually reduced, which indicates that it is more difficult to get the user's real location from the anonymous set. The DLS scheme has the highest probability of query recognition under the same attack. The reason is that once the history query probabilities of two locations are different in the DLS scheme, their chosen optimal dummy location sets must be different. In this property, an adversary can easily identify the user's real location from an anonymous set. We can also see that the probability of query recognition of the Enhanced-DLP and the DLP scheme is the lowest. Although the query recognition probability of the DLP scheme decreases with the increase of $k$, the probability of query recognition is still higher than that of the Enhanced-DLP scheme. Moreover, the probability of query recognition of the Enhanced-DLP is 33.63% lower than that of the DLP scheme. The probability of query recognition of the Enhanced-DLP scheme does not vary much with the increase of anonymity degree $k$. This is because the Enhanced-DLP scheme adopts an enhanced greedy scheme to successfully select dummy locations which ensures the entropy of the current anonymous set is the largest, while the DLP scheme adopts the greedy scheme to select dummy locations using the enumeration scheme from the candidate set. This is vulnerable to privacy breaches from ADLS attacks and requires more time to generate anonymous sets.

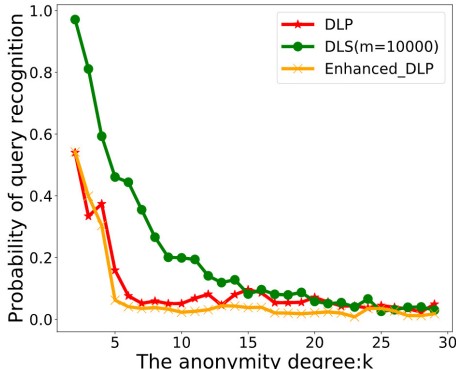

**Figure 5.** Scenario 2: The probability of query recognition versus $k$.

## 5. Related Work

### 5.1. Privacy Protection and Security in the Internet of Things

In recent years, many researchers have proposed some privacy protection schemes in IoT services and applications. In the cloud services of the Internet of Things, Henze et al. [15] proposed a user-driven privacy enhancement scheme that focuses on protecting privacy for individual end-users. Appavoo et al. [16] proposed a lightweight privacy protection model to reduce privacy threats when using untrusted third-party services providers, which may disclose users' information to third parties. Jin et al. [17] introduced a framework for a smart city implemented through the Internet of Things, which encompasses the entire urban information system and forms the transformation of the existing physical system network. Lai et al. proposed a conditional privacy protection authentication with access link (CPAL) [18] in the roaming service to provide users with universal secure roaming services and multi-level privacy protection. Premnath et al. [19] evaluated the cost of breaking the public key cryptosystem when the attacker is limited by the available resources and time, and then the author presented the trade-off between the processing load of the IoT node and the expected privacy protection. Premnath et al. [19] estimated the expense of cracking the public key cryptosystem when the attacker is limited by available resources and time. The author then described the trade-off between the processing load of the IoT node and the expected privacy protection time. Wu et al. [20] introduced a continuous location entropy and trajectory entropy based on the gravity mobility model in the Internet of Things. Yin et al. [21] proposed a location privacy protection scheme that satisfies the differential privacy constraint to protect the user's location information privacy, maximizing data availability in the industrial Internet of Things. Wu et al. [22] introduced a privacy protection scheme to solve the privacy protection problem in the Internet of Things, named EUROCRYPT15. Based on the ITS background, a privacy protection solution was proposed in [23] that relies on a game theory model for private concessions or active attacks. He et al. [24] analyzed the importance of source nodes in WSN networks and proposed a sector-based random routing (SRR) protocol for WSNs to protect source location privacy and balance energy consumption. It is extremely critical to protect the devices and sources in the IoT ecosystem. Lekidis et al. [25] investigated the main trends of IoT access control and the existing authorization framework and summarized the main requirements and evaluation criteria of IoT authorization frameworks. Venckauskas et al. [26] described a modeling framework that models the main attributes of an energy–security–environment problem to define the quality of service for IoT-based applications. Zhang et al. [27] proposed a multi-access edge computing(MEC) enhanced mobility support system (MSS) that uses MEC to reduce operating costs and overhead, and protect the location privacy of IoT users when uploading data and transmitting control information.

### 5.2. LBS Location Anonymous Scheme

The location anonymous scheme is one of the most important techniques for protecting location privacy, and it attempts to make the user's location indistinguishable from other users. The common schemes for protecting a user's location privacy are spatial-temporal cloaking and location obfuscation. K-anonymity is one of the important techniques for location anonymization, which needs to rely on a trusted third-party anonymous server to extend the location of user queries to a larger hidden area, namely the cloaking region (CR) to cover other k-1 users. Gedik et al. [28] proposed a personalized k-anonymity model which enabled users to have different privacy requirements in different contexts and for different users to have different levels in the same context. An scheme based on k-anonymity and l-diversity was proposed in [29], which can ensure that CR has at least k vehicles (k-anonymous) and road segments (l-diversity) when constructing an invisible area, which can effectively protect the user's location privacy. Hossain et al. [30] proposed a weighted adjacency graph based on the k-anonymous technique, the scheme can support k-nearest neighbor queries without revealing the personal information of the query initiator. Moreover, it not only ensures user privacy but also reduces bandwidth usage. Zhang et al. [13] proposed a context-aware location privacy-preserving solution with

differential perturbation, which can enhance the user's location privacy without requiring a trusted third party (TTP). Zhang et al. [31] proposed a scheme of enhancing location privacy schemes which transformed the user-defined grid into a unified grid and adopted both order-preserving symmetric encryption (OPSE) and the k-anonymity technique to protect the user's location privacy, in which the cache technology was used to reduce the overhead of the LBS server. Sun et al. [32], introduced the user's query range to present a novel anonymous region construction scheme. Wu et al. [33] proposed a smart scheme named BUSA to approach the tension between privacy protection and recommendation accuracy. A dual privacy preserving (DDP) [34] scheme was proposed to protect the user's trajectory and query privacy in continuous LBS effectively and to reduce the computation and communication overhead of the single anonymizer.

### 5.3. Dummy-Based Location Selection in the Internet of Things Privacy Protection

The main purpose of the dummy-based location selection scheme is to select a dummy location for the anonymous user. Niu et al. [8] proposed a dummy-location selection (DLS) scheme achieving k-anonymity for LBS users. However, the computation cost of the scheme is very high. Based on the entropy metric [11], the DLS scheme considers the side information which may be exploited by the adversary to select the dummy location. To ensure that the chosen dummy locations can spread as far as possible, the authors also proposed an enhanced-DLS [8] scheme that extends the clocking region while maintaining the same level of privacy as the DLS scheme. Lu et al. [14] considered the need for privacy zones and proposed two schemes of virtual generation: circle-based and grid-based dummy. The authors analyzed the existing problems of the DLS scheme and designed an ADLS attack scheme, but the DLS could not resist the attack of the ADLS scheme [9]. Shaham et al. [35] incorporated a new type of side information and presented a new metric named transition-entropy. Then, they designed an scheme called robust dummy generation (RDG) which can resist the Viterbi attack. Sun et al. [6] proposed a k-anonymity based scheme which preserves user location information by generating dummy locations that can be used to hide the user's real location. At present, most of the existing schemes utilize dummy location to protect the user's location privacy. However, the computation complexity of the schemes are still very high. Therefore, how to effectively select the dummy location in the Internet of Things is still a challenge. In this paper, we introduced the Enhanced-DLP for IoT user and devices. By considering side information and computational overhead, the Enhanced-DLP can effectively select the reasonable dummy locations and protect against side information attacks.

## 6. Discussion

### 6.1. Limitations of Our Scheme

If there is a location that has never been visited, its historical query probability is 0%, and the location where the historical query probability is zero contains some desert, oceans and some areas that are not suitable for human survival. A dummy location generated may be invalid dummy location, which may expose the user's location information.

The dataset we used in the lab is relatively small, and the run time obtained by the scheme is short. However, if there are massive locations in the real environment data set, the run time of the enhanced-DLP may be different, thus affecting the efficiency of the scheme to generate dummy locations.

### 6.2. Threats-to-Validity

We can see from the experimental results, that the probability of the scheme disclosing the user's location is low, as the dummy location we generated is as close as possible to the user's location, which makes it hard for the attacker to distinguish which one is the user's real location. Suppose $u_1, u_2, u_3, ..., u_n$ are locations and $u_i$ is the user's location. In the first case, the number of locations with the same query probability as the user's location is greater than k. Based on such conditions,

the enhanced-DLP scheme obtains the historical query probability of the dummy locations in the anonymous set as the same, and we get an anonymous set $(u_1, u_i, u_3, u_4, u_5)$, therefore, the attacker can not get the user's location information. In the second case, the number of the historical query probability of the user location is the same as the historical query probabilities of other locations (not less than 1) and an anonymous set $(u_4, u_6, u_9, u_i, u_{10})$ is obtained by the enhanced-DLP scheme. This anonymous set guarantees a larger entropy approximate to the optimal value, thus the attacker cannot recognize the user's location. In the third case, if there is no location similar to the user's real location. The enhanced-DLP scheme randomly chooses one location from the candidate set and adds to the anonymous set, and then uses the greedy scheme to select the remaining k-1 dummy locations, which changes the regular pattern of the location in the anonymous set. It is guaranteed that the anonymous set contains the maximum entropy, and thus the attacker cannot easily recognize the user's location.

## 7. Conclusions

In this paper, we present an Enhanced-DLP scheme for IoT services and applications to protect the user's location privacy. In the Enhanced-DLP scheme, we adopted the enhanced greedy scheme instead of the original greedy scheme, making full use of the relationship between the data, so that the optimal dummy location can be found easily without traversing all the locations in the candidate set. The Enhanced-DLP scheme can guarantee the same entropy as the anonymous set generated by the existing scheme. In the verification of the effectiveness of the Enhanced-DLP scheme, we conduct the simulations on two scenarios. In the first scenario, the DLP and the Enhanced-DLP use almost the same scheme when generating the dummy location, so the computational cost of the two schemes are almost the same. In the second scenario, the dummy location generation time of the Enhanced-DLP scheme is 68.14% lower than that of the DLP scheme, and the probability of the Enhanced-DLP scheme against the ADLS scheme is 33.63% stronger than that of the DLP scheme. Our scheme only considers the query probability as the side information and the ADLS scheme as the attack scheme.However, but in such side information, the user's location information cannot be completely hidden. In subsequent research, we will consider more background knowledge and context in the process of generating dummy locations, and more attack schemes will be adopted.

**Author Contributions:** Conceptualization, G.C., X.Z. and Y.D.; methodology, G.C., T.L. and X.Z.; software, G.C.; validation, G.C. and X.Z.; formal analysis, G.C., X.Z. and T.L.; investigation, G.C.; resources, Y.D.; data curation, G.C.; writing—original draft preparation, G.C.; writing—review and editing, G.C., X.Z., Y.D., T.L. and J.J.; visualization, G.C.

**Funding:** This research was funded by name of the NSFC project (grant no. 61762058, and no. 61861024), the Science and Technology project of Gansu Province (grant no. 1610RJZA056), the youth Science Foundation of Lanzhou Jiaotong University (grant no. 2014026), Foundation of A Hundred Youth Talents Training Program of Lanzhou Jiaotong University and the National Key Research and Development Program of China(2017YFB0802201).

**Conflicts of Interest:** The authors declare no conflict of interest.

## Abbreviations

The following abbreviations are used in this manuscript:

| | |
|---|---|
| IoT | Internet of Things |
| POI | point of interest |
| LBS | Location-based services |
| LSP | Location-based services providers |
| DLS | Dummy location selection |
| DLP | dummy-based location privacy-preserving |
| ADLS | Attack algorithm for DLS |
| Enhanced-DLP | Enhanced dummy-based location privacy-preserving |

| CPAL | conditional privacy protection authentication with access link |
| SRR | sector-based random routing |
| MEC | multi-access edge computing |
| MSS | mobility support system |
| CR | cloaking region |
| TTP | trusted third party |
| DDP | dual privacy preserving |
| RDG | robust dummy generation |

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
