# Peer review of "An Efficient Dummy-Based Location Privacy-Preserving Scheme for Internet of Things Services"

_information, doi:10.3390/info10090278_

Round 1

Reviewer 1 Report

The work is well structured however there are several typos that must be addressed.

A non-exhaustive list is the following

part 53: "dummy locations are selected build"

part 62: "set. it can"

part 75: "user?s"

part 101: "the Enhanced-DLP need to"

part 101: "and puts the location"

part 113: "and the randomly"

part 150: "k locations by randomly selecting."

part 152: "locations. we can"

part 184: "locations for anonymous."

Apart from the presentation issues, the authors should consider resolving the following:

The introduction or preliminaries section needs to be expanded to include a more thorough description of the DLP algorithm and an algorithmic comparison of the proposed extension with the standard DLP emphasizing/explaining the greedy nature of the algorithm.

A better explanation of the concept of dummy locations should be added.

According to line 219: "complexity of the DLP algorithm (i.e., O(n^2)) is higher than the Enhanced-DLP algorithm(i.e., O(n))." However, this is not apparent in Figure 3. In the same context, what is n? It appears that the authors do not imply k. In this case Figure 3 should be changed accordingly.

Author Response

Reply to Review Reports

Dear editor,

  Thanks for the reviewers for giving us useful suggestions which would help us both in English and in depth to improve the quality of our manuscript. Here we submit our revised manuscript with the title “Efficient Dummy Location Selection Algorithm for Internet of Things Services”, which has been modified according to reviewers’ suggestions. A great amount of efforts was also made to correct the mistakes and improve the quality of the manuscript. We highlight all the changes in the revised manuscript.

Sincerely yours,

Gang Cai

The following is a point-to-point response to the three reviewer’s comments.

Reviewer #1:

The work is well structured however there are several typos that must be addressed.

Answer: Thanks for the reviewer’s comments. All typing and grammatical errors that exist in the article are modified as much as possible and marked in red.

The introduction or preliminaries section needs to be expanded to include a more thorough description of the DLP algorithm and an algorithmic comparison of the proposed extension with the standard DLP emphasizing/explaining the greedy nature of the algorithm.

Answer: Thanks. The description of the DLP algorithm is given in Section 2.3, and the nature of the greedy algorithm is given (line 101-lilne 108). At line108-line116, we summarize the differences between the Enhanced-DLP and the standard DLP algorithm.

A better explanation of the concept of dummy locations should be added.

Answer: Thanks. The detail explanation of dummy locations is given in line38-line40.

According to line 219: "complexity of the DLP algorithm (i.e., O(n^2)) is higher than the Enhanced-DLP algorithm (i.e., O(n))." However, this is not apparent in Figure 3. In the same context, what is n? It appears that the authors do not imply k. In this case Figure 3 should be changed accordingly.

Answer: Thanks. In Figure 3, since both Enhanced-DLP and DLP can only use the same optimal processing method under such dataset (if kb>k), which can't reflect our modification of DLP algorithm. But we can see it from Figure 4. Using n in time complexity is just an idiomatic usage. In fact, our n and k are the same. We explained this in line 276.

In conclusion, I suggest this paper to be accepted with a minor revision.

Reviewer 2 Report

The paper proposed a lightweight dummy location privacy-preserving scheme for Internet-of-Things (IoT). The experimental results illustrate that the method can be applied to protect the user’s location privacy in IoT applications and services. The paper is interesting and relevant; however the following issues should be addressed before the paper could be accepted:

1. The novelty and scientific contribution of the paper should be explicitly formulated in the Introduction Section.

2. Figure captions should be more informative and descriptive.

3. Figure 2: the model visualization is not clear. Specifically, should not the user‘s location originate outside of the user‘s device?

4. The security analysis in subsection 3.2 is incomplete. Specifically, what about false data injection attacks?

5. Figures 3, 4, part a): I see the results of three methods, not five as presented in the legend.

6. Figures 3, 4, part b): figure should presents and improvement in performance over the baseline method rather than absolute time.

7. Section 5.1: The overview of related works misses several important studies on the topic of IoT security and privacy protection. The following works are suggested to be discussed and cited:

* Access control in Internet-of-Things: A survey. Journal of Network and Computer Applications, 144, 79–101. doi:10.1016/j.jnca.2019.06.017

* Multi-access edge computing aided mobility for privacy protection in Internet of Things. Computing, 101(7), 729–742. doi:10.1007/s00607-018-0639-0

* A sector-based random routing scheme for protecting the source location privacy in WSNs for the Internet of Things. Future Generation Computer Systems, 96, 438–448. doi:10.1016/j.future.2019.02.049

* Modelling of Internet of Things units for estimating security-energy-performance relationships for quality of service and environment awareness. Security and Communication Networks, 9(16), 3324–3339. doi:10.1002/sec.1537

8. Add a discussion section. Discuss limitations of the method and threats-to-validity.

9. Conclusions section should be rewritten to include main numerical results and implications for future research.

Author Response

Reply to Review Reports

Dear editor,

  Thanks for the reviewers for giving us useful suggestions which would help us both in English and in depth to improve the quality of our manuscript. Here we submit our revised manuscript with the title “Efficient Dummy Location Selection Algorithm for Internet of Things Services”, which has been modified according to reviewers’ suggestions. A great amount of efforts was also made to correct the mistakes and improve the quality of the manuscript. We highlight all the changes in the revised manuscript.

Sincerely yours,

Gang Cai

The following is a point-to-point response to the three reviewer’s comments.

Reviewer: #2

The paper proposed a lightweight dummy location privacy-preserving scheme for Internet-of-Things (IoT). The experimental results illustrate that the method can be applied to protect the user’s location privacy in IoT applications and services. The paper is interesting and relevant; However, the following issues should be addressed before the paper could be accepted:

The novelty and scientific contribution of the paper should be explicitly formulated in the Introduction Section.

Answer: Thanks for your comments. The novelty and scientific contribution of the paper you mentioned in the paper I have listed in the first section of the paper. Please see lines 63 to 71 for details.

Figure captions should be more informative and descriptive.

Answer: Thanks for your comments. I have add more information for Figure captions and more description. Please look at the bottom of the picture, and line 279-line 295 for details.

Figure 2: the model visualization is not clear. Specifically, should not the user’s location originate outside of the user‘s device?

Answer: Thanks for your comments. I have re-examined and drawn the system model diagram.

The security analysis in subsection 3.2 is incomplete. Specifically, what about false data injection attacks?

Answer: Thanks for your comments. I think the security analysis is complete, and other types of attacks can be summarized into these two categories. For example, the false data injection attacks you mentioned can be considered an aspect of a collusion attack.

Figures 3, 4, part a): I see the results of three methods, not five as presented in the legend.

Answer: Thanks for your comments. Figures 3 and 4, part a, there are four algorithms get the same entropy, so their images overlap, allowing you to see the results of only three algorithms. So, I redraw the result graph of the six algorithms, so that the effect of the algorithm can be clearly displayed.

Figures 3, 4, part b): figure should present and improvement in performance over the baseline method rather than absolute time.

Answer: Thanks for your comments. In Figures 3 and 4, part b, we can clearly see the difference between the two algorithm and the features of the two algorithms, and can see the trend of the two algorithms. In order to better display the results of the experiment, I used the polyline to draw the experimental results.

Section 5.1: The overview of related works misses several important studies on the topic of IoT security and privacy protection.

Answer: Thanks for your comments. The missing IoT security and privacy protection in related work has been supplemented in the paper. Please see line 320–line 329 in 5.1 subsection and reference [23], [24], [25], [26] for details.

Add a discussion section. Discuss limitations of the method and threats-to-validity.

Answer: Thanks for your comments. A new section on the discussion of some issues is available in the article, please see section 6 for details.

Conclusions section should be rewritten to include main numerical results and implications for future research.

Answer: Thanks for your comments. The summary in the paper has been rewritten and gives some directions for future research. please see section 7 for details.

In conclusion, I suggest this paper to be accepted with a minor revision.

Reviewer 3 Report

The paper is interesting and proposes a new method for improving location privacy in scenarios is which the location of an end user has to be evaluated for further analyses.

Anyway, the paper requires an overall revision, since it contains some typos and some parts can be improved.

Some examples (on the overall required modifications):

In the last paragraph of the Introduction there are some missing blanks and a sentence that starts but not ends in the proper way. Figure --> Fig. Formula --> Eq. Please detail why in Eq. 2 authors choose the log_2 instead of log_10. Subsection 2.3 has to be re-styled, together with the images, which are not in a sufficient quality and dimension (and, sometimes, they contains some typos). There are many "?" marks in place of "'". ADLS acronym is not defined in its extended version. Subsection 3.1 has to be revised, since some notations are defined but not used in the right way after their definition, and detail in which way the "n" locations specified before Eq. 3 are correlated with the index "i" defined in the beginning of the Subsection itself. Please move the website URL from row 181 to the References section. There are missing blanks in rows 186-187. Fig. 3-5 should be re-styled and enlarged, since they are a bit unreadable for a reader.

Author Response

Reply to Review Reports

Dear editor,

  Thanks for the reviewers for giving us useful suggestions which would help us both in English and in depth to improve the quality of our manuscript. Here we submit our revised manuscript with the title “Efficient Dummy Location Selection Algorithm for Internet of Things Services”, which has been modified according to reviewers’ suggestions. A great amount of efforts was also made to correct the mistakes and improve the quality of the manuscript. We highlight all the changes in the revised manuscript.

Sincerely yours,

Gang Cai

The following is a point-to-point response to the three reviewer’s comments.

Reviewer #3:

In the last paragraph of the Introduction there are some missing blanks and a sentence that starts but not ends in the proper way. Figure --> Fig. Formula --> Eq.

Answer: Thanks for your comments. The above-mentioned problems that you mentioned out in the paper have been revised. For details, please see the high light of the paper.

Please detail why in Eq. 2 authors choose the log_2 instead of log_10.

Answer: Thanks for your comments. Because of the representations used in some of the previous studies[8], [9], I followed the results of my predecessors and made no changes. Please see reference [8], [9] for details

Subsection 2.3 has to be re-styled, together with the images, which are not in a sufficient quality and dimension (and, sometimes, they contain some typos).

Answer: Thanks. There are some problems in the image in subsection 2.3, and the some mistakes in the expression. I have already revised the subsection 2.3. Please see subsection 2.4 for details.

 There are many "?" marks in place of "'". ADLS acronym is not defined in its extended version.

Answer: Thanks. I have already revised the mistakes you mentioned and added the ADLS acronym in line 426.

Subsection 3.1 has to be revised, since some notations are defined but not used in the right way after their definition, and detail in which way the "n" locations specified before Eq. 3 are correlated with the index "i" defined in the beginning of the Subsection itself.

Answer: Thanks for your comments. There are some problems what you mentioned above, I have made a revision. Please seen subsection3.1 for details.

Please move the website URL from row 181 to the References section. There are missing blanks in rows 186-187. Fig. 3-5 should be re-styled and enlarged, since they are a bit unreadable for a reader.

Answer: Thanks for your comments. I have already made a revision of the problem that you mentioned in the paper. Please see the paper for details.

In conclusion, I suggest this paper to be accepted with a minor revision.

Round 2

Reviewer 2 Report

I am fully satisfied with the revisions of the paper. The paper has been improved significantly. I have no further comments and recommend the paper to be accepted.